# Phaeohyphomycosis in Solid Organ Transplant Recipients: A Case Series and Narrative Review of the Literature

**DOI:** 10.3390/jof9030283

**Published:** 2023-02-21

**Authors:** Davide Lo Porto, Andrea Cona, Francesca Todaro, Elena De Carolis, Francesca Cardinale, Neha Hafeez, Giuseppina Di Martino, Pier Giulio Conaldi, Maurizio Sanguinetti, Paolo Antonio Grossi, Alessandra Mularoni

**Affiliations:** 1Unit of Infectious Diseases, ISMETT-IRCCS Istituto Mediterraneo per i Trapianti e Terapie ad Alta Specializzazione, Via E. Tricomi, 5, 90127 Palermo, Italy; 2Dipartimento di Scienze di Laboratorio e Infettivologiche, Fondazione Policlinico Universitario A. Gemelli IRCCS, 00168 Rome, Italy; 3Department of Medicine, University of Pittsburgh School of Medicine, Pittsburgh, PA 15261, USA; 4Infectious and Tropical Diseases Unit, Department of Medicine and Surgery, University of Insubria-ASST-Sette Laghi, 21100 Varese, Italy

**Keywords:** *Alternaria alternata*, *Alternaria infectoria*, *Curvularia hawaiiensis*, phaeohyphomycosis, SOT, voriconazole, itraconazole, posaconazole, transplantation

## Abstract

Phaeohyphomycosis comprises a variety of infections caused by pigmented fungi. Solid organ transplant (SOT) recipients are particularly at risk of invasive infections due to their prolonged immunosuppression. Here, we describe three cases of phaeohyphomycosis in SOT recipients who were successfully treated with surgical excision and/or antifungal therapy. We additionally carried out a narrative review of the literature on phaeohyphomycosis in 94 SOT recipients from 66 published studies describing 40 different species of fungi. The most reported fungus was *Alternaria* (21%). The median time from transplant to diagnosis was 18 months (IQR 8.25–48), and kidney transplants were the most reported. Antifungal regimens were not homogeneous, though there was a prevalence of itraconazole- and voriconazole-based treatments. Clinical outcomes included recovery in 81% and death in 5% of infected SOT recipients. Susceptibility testing was done in 26.6% of the cases, with heterogeneous results due to the variety of species isolated. While the wide diversity of dematiaceous fungi and their host range make it difficult to offer a uniform approach for phaeohyphomycosis, an early diagnosis and therapy are critical in preventing the dissemination of disease in the immunocompromised host.

## 1. Introduction

The term “phaeohyphomycosis” refers to subcutaneous, superficial, and systemic infections caused by pigmented (dematiaceous) fungi. There are over 100 species, and over 50 genera of these fungi, which have been associated with human disease [1]. The presence of melanin in their cell walls is characteristic of the condition, and likely an important virulence factor, believed to be involved in evasion from the host immune response [2]. Most human diseases caused by these fungi are noninvasive (e.g., onychomycosis, superficial cutaneous infection, chronic allergic fungal sinusitis) and associated with low mortality. However, rarely, phaeohyphomycosis can lead to invasive complications, including skin and subcutaneous disease, pneumonia, central nervous system (CNS) disease, fungemia, and multi-organ disseminated disease. Solid organ transplant (SOT) recipients are particularly at risk of invasive infections due to prolonged immunosuppression [3,4]. We describe three cases of phaeohyphomycosis caused by *Alternaria alternata*, *Curvularia hawaiiensis*, and *Alternaria infectoria* in two heart recipients and one kidney recipient, respectively. Additionally, we provide a narrative review of the literature on phaeohyphomycosis infections occurring in SOT recipients over the last 10 years.

## 2. Materials and Methods

### 2.1. Case Series

We report three cases of SOT recipients diagnosed with cutaneous and subcutaneous phaeohyphomycosis caused by *Alternaria alternata, Alternaria infectoria*, and *Curvularia hawaiiensis*. The identification of the species was made using both traditional microbiology (on a sabouraud chloramphenicol gentamicin agar growth medium) and molecular biology by isolating 18S rRNA, a small subunit ribosomal ribonucleic acid (SSU rRNA), a component of the eukaryotic ribosomal small subunit (40S). All patients provided written informed consent for this research.

### 2.2. Review of the Literature

We carried out a narrative review of the literature using PubMed with the following search terms: “phaeohyphomycosis” AND “transplant,” selecting only SOT recipients. All relevant articles from January 2011 to August 2022 were screened, selecting cases diagnosed between January 2011 and August 2022. Articles published in English, Spanish, and French were included. The cases were divided into those reported in Europe and cases from other continents. The following data were included: the type of transplant, time from transplant to diagnosis, localization of the infection, the species of fungi, susceptibility test availability, type of treatment, and clinical outcome. We defined “full recovery” for patients who were described as completely cured after therapy and “partial recovery” when there was an improvement but no clinical cure. We considered cases of phaeohyphomycosis to be “local infections” if they involved only the skin and subcutaneous tissues. We considered cases to be “local deep” if they were localized in deep tissues such as bones, glands, or mucous membranes. Finally, infections were classified as “disseminated” when they involved internal organs, such as the brain, liver, lungs, or non-contiguous bones [4].

## 3. Results

### 3.1. Case 1

A 53-year-old male presented with a cutaneous lesion on the right leg 7 months after a heart transplant was performed for cardiac failure due to giant cell myocarditis. His clinical history was marked by acute cellular rejection at month 1 after the transplant, treated with steroid boluses. At the time of the skin lesion appearance, his immunosuppressive therapy was based on 15 mg of prednisone once daily, 750 mg of mycophenolate mofetil (MMF) twice daily, and tacrolimus (FK) with a trough level of around 14 ng/mL. The cutaneous lesion presented as a hard ulcerated nodule 1 cm in diameter draining purulent fluid (Figure 1A). The lesion was scraped by the Infectious Disease and Dermatology team and was cultured. Two weeks after the mycelium was grown at 30 °C on a sabouraud chloramphenicol gentamicin agar plate, a scraping culture was made and after a microscopic examination, DNA extraction was performed. A polymerase chain reaction using 18S r-DNA revealed *Curvularia hawaiiensis* and a topical therapy with bifonazole was started. The strain susceptibility assay was performed at the reference center (Università Cattolica-Policlinico Gemelli) with Sensititre™ YeastOne ITAMYUCC, (Thermo Fisher Scientific, Cleveland, OH, USA) broth micro-dilution, following CLSI methods for filamentous fungi [5] (Table 1). ESCMID and ECMM joint clinical guidelines were used for the results interpretation [6].

After 4 weeks of topical therapy, there was no improvement in the cutaneous lesion, and the patient developed eyelid ptosis, so a total-body CT scan was performed over concern for cerebral dissemination. There was no evidence of cerebral involvement, but two undefined pulmonary nodules were noted in the right lung and one in the left. The cutaneous lesion was surgically removed, and a systemic therapy with 400 mg of itraconazole once daily commenced. The patient was informed to take itraconazole capsules with food and acid drinks to obtain adequate plasma concentrations. FK dosing was reduced by 60% with monitoring of the trough levels. The therapeutic drug monitoring of itraconazole, measured by HPLC, showed a trough level of 1.4 mg/L and a peak level of 1.9 mg/L; the administered dose was unchanged for the duration of the therapy. Two months later, there was no evidence of recurrence of the cutaneous lesions, and the pulmonary nodules at CT were unchanged. The therapy was administered for 4 months. At the 5-month follow-up the patient was stable, with no recurrence.

### 3.2. Case 2

A 63-year-old male, working as a tomato grower, presented with a cutaneous erythematous nodular lesion 8 months after a heart transplant. His immunosuppressive therapy consisted of 20 mg of prednisone daily, 500 mg of MMF twice daily, and tacrolimus with a trough level of around 8 ng/mL. His medical history included an acute cellular rejection, treated with steroid boluses at 2 months after the transplant.

The lesion, on his left thigh, was 1.5 cm in diameter with a crusty and ulcerated surface that had been growing in dimension over the previous 3 months, followed by 2 satellite lesions on the same limb (Figure 1B). No pain or itching was present. The lesions were concerning for of a lymphoproliferative malignancy; as a result, an excisional biopsy was done. The histopathological examination showed no sign of malignancy, but a significant inflammatory pattern was found, with neutrophilic infiltrate and the presence of hyphae and yeast-like structures. Molecular biology (18s-rRNA) performed on the excisional biopsy from the lesions on the thigh identified *Alternaria alternata.*

A chest and brain CT scan was done, though no sign of dissemination was found. After the excisional biopsy, an oral therapy with isavuconazole was administered for 2 months, monitoring the FK trough levels. Four months after the cessation of therapy, the patient presented with two new cutaneous lesions on his forearms, which were surgically removed. A new histopathological examination found the same pattern as the first lesion, and *Alternaria alternata* was identified again with molecular biology and culture (Figure 2 and Figure 3). The fungal isolate was sent for susceptibility testing to the reference center (Università Cattolica-Policlinico Gemelli). Based on the result of the susceptibility test [5] (Table 2), therapy with itraconazole commenced. Therapy was administered after meals, with lemon juice to improve the absorption. Monitoring of the trough level of itraconazole was done 7 days after starting the therapy, showing a trough level of 0.4 mg/L, which was under the therapeutic range (target range level between 0.5 and 1 mg/L), and the patient was told to take itraconazole 12 h apart from the therapy with the omeprazole he was taking simultaneously. The tacrolimus dose was reduced by 60%. Itraconazole was administered for 3 months. At the 6-month follow-up, there was no sign of new lesions or of dissemination.

### 3.3. Case 3

A 59-year-old male, working as a wheat farmer, with a history of kidney transplant for an IgA nephropathy performed 10 years prior, presented for a routine dermatological consult. His immunosuppressive therapy consisted of 500 mg of MMF twice daily, 2.5 mg of prednisone daily, and FK with a trough level of 6 ng/mL. A physical examination showed a lesion behind the ear and two other lesions, one on each knee (Figure 1C). The lesions on the knees were two crusty painless nodules that had been growing for several months. The lesion behind the ears was surgically removed, and the histological examination showed pigmented basal cell carcinoma. Scraping was done from the knee lesions, which yielded *Alternaria infectoria* as the species responsible for the infection. A total-body CT scan was done, and no other lesions were found. The lesions on the knees were surgically removed, and an inflammatory pattern was found at histological examination. No antifungal systemic therapy was administered, and no signs of relapse were found at the one-year follow-up.

### 3.4. Literature Review

A narrative review of the literature yielded a total of 66 published studies, with 94 cases from several countries (Argentina, Australia, Austria, Belgium, Brazil, China, Colombia, Czech Republic, France, French Antilles, Germany, Japan, India, Italy, Kuwait, Portugal, Singapore, Slovakia, Spain, Thailand, and the U.S.). We divided the cases into those reported from Europe (Eu *n* = 36) and from other continents (Non-Eu *n* = 58) (Table 2). Among studies published outside Europe, 28 reports were from America, 26 were from Asia, and 4 were from Australia. Most of the cases (75%) were published between 2016 and 2022. A geographical distribution was noted, with 85% of the cases reported from countries with a mild or tropical climate. The majority of the cases in the EU were reported from Spain and France (Table 2, Table 3, Table 4, Table 5, Table 6 and Table 7).

### 3.5. Reported Cases from Europe vs. Outside Europe

Our literature review revealed a total of 94 cases, with 38% reported in the EU and 62% reported outside the EU. In total, 73% of the infections were reported in kidney transplants recipients, 14% were in lung transplants, 7.5% were in heart transplants, and 5.5% were in liver transplants. When comparing the EU and non-EU cases, lung transplant represented 36% and 0%, kidney transplant 45% and 91%, heart transplant 11% and 5%, and liver transplant 8% and 4%, respectively. Overall, the median age of patients at disease presentation was 56 years old (54 years old in EU vs. 58.5 in non-EU) and a male/female ratio of 3.5 in EU vs. 2.3 in other countries. The median time from transplant to diagnosis in months was 18, with no relevant differences all over the world. From all the reported cases, 70% were classified as “local,” 22% as “disseminated,” and 8% as “local deep”, with no differences between EU and non-EU countries.

We found approximately 40 different species of fungi causing phaeohyphomycosis in our review. Among cases with available pathogen identification, the most reported genus was *Alternaria* (19/81, 24%), followed by *Exophiala* (13/81, 16%) worldwide. In Europe, the causative microorganism was available in all of the cases; we found 16 different species of fungi, in most of the cases belonging to the genus *Alternaria* (44% of European cases), though *Cladophialophora bantiana, Exophiala spp,* and *Medicopsis romeroi* were commonly reported. Outside of Europe, in 52% of the reported cases, the identification of the pathogen was not available. In the remaining cases, over 30 different species of fungi were identified. Most cases involved the genus *Exophiala* (10/45, 22% of non-EU cases), though *Alternaria* spp. was also frequently reported. The identification of the pathogen was made using cultures in about 90% of the total cases (89% EU vs. 91% non-EU), while the use of molecular biology was reported in 58% of the cases (78% EU vs. 45% non-EU). In the remaining cases, an identification was made by direct microscopy or histological examination.

Susceptibility testing was done in 26.6% of the total cases. Susceptibilities to antifungal agents and treatment regimens were quite heterogeneous due to the variety of isolated species. Surgery was performed in 68% of total cases. In total, 10% of patients were treated with surgery alone, with no antifungal therapy, though this approach was more commonly reported in Europe (in 17% of EU cases vs. 5% of non-EU cases). The most used antifungal agents were voriconazole in Europe and itraconazole outside of Europe, followed by posaconazole and amphotericin B. The majority of patients (67%) were treated with a single antifungal agent (58% rate of monotherapy in EU vs. 72% in non-EU), and the rest with more than one agent, in sequential or combination therapy. The median length of antifungal therapy was 16 weeks, but in 41.5% of reports, the duration of therapy was not specified. Regarding the clinical outcome, recovery was reported in 81% of cases, with no relevant differences between EU (81%) and non-EU (85%). Five percent of patients died due to the fungal infection.

**Table 3 jof-09-00283-t003:** Reported cases in Europe.

Year	Ref	Country	Sex	Age	Dissemination	Location	Species	Method of Identification	Therapy	Duration of Therapy in Weeks	Surgery	Transplant	Outcome	Time Tx-Diagnosis in Months	Susceptibility Test
2019	[7]	Spain	M	58	Local	Skin, foot	*Alternaria alternata*	Culture + molecular biology (18s-DNA, ITS 1 + ITS 4)	ITZ + VZL + TBF	8	No	Lung	FR	42	None
2019	[7]	Spain	M	68	Local deep	Foot tendon	*Alternaria alternata*	Culture + molecular biology (18s-DNA, ITS 1 + ITS 4)	Surgery	-	Yes	Lung	FR	48	None
2019	[7]	Spain	M	32	Local	Skin, legs, and wrist	*Alternaria alternata*	Culture + molecular biology (18s-DNA, ITS 1 + ITS 4)	ITZ	24	Yes	Lung	FR	3	None
2015	[8]	Portugal	M	65	Local	Skin, hand, and leg	*Alternaria alternata*	Culture + molecular biology (ITS 1 + ITS 4)	ITZ	12	Yes	Liver	FR	6	None
2019	[9]	Italy	M	68	Local	Skin, hand	*Alternaria alternata*	Culture + molecular biology	ISZ + PZL	-	No	Kidney	FR	48	Yes
2020	[10]	Italy	F	56	Local	Skin, limbs	*Alternaria alternata*	Culture	VZL	24	No	Liver	FR	108	Yes
2019	[7]	Spain	F	53	Local	Skin, leg	*Alternaria infectoria*	Culture + molecular biology (18s-DNA, ITS 1 + ITS 4)	ITZ	28	No	Lung	FR	24	None
2019	[7]	Spain	M	64	Local	Skin, legs	*Alternaria infectoria*	Culture + molecular biology (18s-DNA, ITS 1 + ITS 4)	ITZ + VZL	72	No	Lung	FR	2	None
2019	[7]	Spain	M	51	Local	Skin, leg	*Alternaria infectoria*	Culture + molecular biology (18s-DNA, ITS 1 + ITS 4)	Surgery	-	Yes	Kidney	FR	24	None
2019	[7]	Spain	M	56	Local	Skin, leg	*Alternaria infectoria*	Culture + molecular biology (18s-DNA, ITS 1 + ITS 4)	ITZ	8	Yes	Kidney	FR	25	None
2019	[7]	Spain	M	46	Local	Skin, leg	*Alternaria infectoria*	Culture + molecular biology (18s-DNA, ITS 1 + ITS 4)	Topic VZL	8	Yes	Lung	FR	18	None
2017	[11]	Spain	F	72	Local	Skin, leg	*Alternaria infectoria*	Culture + molecular biology (18s + 28s rRNA)	Surgery	-	Yes	Kidney	FR	180	None
2012	[12]	Portugal	M	53	Local	Skin, hands, and feet	*Alternaria infectoria*	Culture + molecular biology (ITS 1 + ITS 4)	ITZ	40	No	Kidney	FR	16	None
2012	[13]	Italy	F	64	Disseminated	Kidney	*Alternaria infectoria*	Undefined	TBF	-	Yes	Kidney	FR	-	Yes
2016	[14]	Czech Republic	M	61	Disseminated	Lungs	*Alternaria infectoria*	Culture + molecular biology (ITS)	VZL + PZL	28	No	Heart	FR	12	Yes
2020	[15]	Spain	M	46	Local	Skin, leg	*Alternaria* spp.	Culture	VZL	3	Yes	Lung	FR	24	None
2014	[16]	Slovakia	M	63	Disseminated	Skin and brain	*Cladophialophora bantiana*	Culture + molecular biology (ITS 1 + ITS 4)	LAB	-	No	Heart	DFI	9	Yes
2017	[17]	France	F	35	Disseminated	CNS, spinal cord, and cerebellum	*Cladophialophora bantiana*	Culture + direct microscopy	LAB + PZL + FTS	36	No	Lung	DOR	120	None
2016	[18]	Belgium	F	34	Disseminated	Bone and brain	*Cladophialophora bantiana*	Culture + molecular biology	VZL + LAB, ISZ + LAB, PZL + LAB + FTS	-	Yes	Kidney	PR	17	Yes
2019	[7]	Spain	M	70	Local	Skin, knee	*Cladosporium cladosporioides*	Culture + molecular biology (18s-DNA, ITS 1 + ITS 4)	Surgery	-	Yes	Kidney	FR	10	None
2013	[19]	Spain	M	25	Local	Skin, leg	*Curvularia lunata*	Culture+ direct microscopy	ITZ	-	Yes	Kidney	FR	18	None
2018	[20]	France	M	51	Local deep	Foot tendon and skin	*Diaporthe raonikayaporum*	Culture + direct microscopy	Surgery	-	Yes	Kidney	FR	84	None
2019	[21]	Austria	M	76	Local deep	Sternal wound	*Exophiala dermatitidis*	Molecular biology (ITS sequence)	FZL + ADF, VZL	-	Yes	Lung	DFI	<1	Yes
2019	[7]	Spain	M	58	Local	Skin, leg	*Exophiala oligosperma*	Culture + molecular biology (18s-DNA, ITS 1 + ITS 4)	VZL	32	Yes	Lung	DOR	20	None
2015	[22]	Italy	F	65	Local	Skin, hand	*Exophiala xenobiotica*	Culture + molecular biology (ITS)	VZL, LAB, PZL	68	Yes	Kidney	FR	18	None
2019	[23]	Spain	M	65	Local	Skin, foot	*Medicopsis romeroi*	Culture + direct microscopy + molecular biology (ITS1 + ITS4)	PZL	4	Yes	Liver	FR	1	Yes
2019	[23]	Spain	F	56	Local	Skin, hand	*Medicopsis romeroi*	Culture + direct microscopy + molecular biology (ITS1 + ITS4)	VZL	10	Yes	Kidney	FR	-	Yes
2020	[24]	France	M	30	Local	Skin, foot	*Medicopsis romeroi*	Culture + molecular biology (ITS 1 + ITS 4)	VZL	8	Yes	Kidney	FR	18	None
2019	[7]	Spain	M	65	Local	Skin, leg	*Microsphaeropsis arundinis*	Culture + molecular biology (18s-DNA, ITS 1 + ITS 4)	ITZ + TBF	54	Yes	Lung	FR	60	None
2015	[25]	France	M	59	Local	Skin, toes, and foot	*Neoscytalidium dimidiatum*	Culture + molecular biology (ITS 1 + ITS 4)	VZL	12	No	Kidney	FR	8	None
2015	[25]	France	M	49	Disseminated	Skin, bones, and lungs	*Neoscytalidium dimidiatum*	Culture + molecular biology (ITS 1 + ITS 4)	VZL	-	No	Kidney	FR	13	Yes
2012	[26]	Germany	M	69	Local	Skin, leg, and abdomen	*Ochroconis Gallopavum*	Culture	VZL	12	Yes	Lung	DOR	72	None
2012	[26]	Germany	M	69	Disseminated	Skin and lung	*Ochroconis Gallopavum*	Culture + direct microscopy	VZL	-	Yes	Lung	DOR	72	None
2012	[27]	France	M	66	Local	Skin, foot	*Pyrenochaeta romeroi*	Molecular biology	Surgery, immunosuppression reduction	-	Yes	Kidney	FR	14	None
2020	[28]	France	M	49	Local	Skin, hand	*Trematosphaeria grisea*	Culture + molecular biology (rDNA 28S D1-D2)	ISZ + TBF	84	No	Heart	FR	3	None
2017	[29]	France	M	71	Local	Skin, leg	*Veronaea botryosa*	Direct microscopy + molecular biology	PZL	12	Yes	Heart	DOR	7	None

PZL = posaconazole. VZL = voriconazole. ITZ = itraconazole. LAB = liposomal amphotericin B. ISZ = isavuconazole. ADF = anidulafungin. CSF = caspofungin. TBF = terbinafine. FTS = flucytosine. FR = full recovery. DOR = death for other reason. DFI = death for fungal infection. ITS = internal transcribed spacer.

**Table 4 jof-09-00283-t004:** Reported cases in the U.S.A.

Year	Ref	Country	Sex	Age	Dissemination	Localization	Species	Method of Identification	Therapy	Duration of Therapy in Weeks	Surgery	Transplant	Outcome	Time Tx-Diagnosis in Months	Susceptibility Test
2020	[30]	U.S.A.	F	54	Disseminated	Brain	*Acrophialophora levis*	Culture + molecular biology (ITS)	VZL	-	No	Kidney	FR	9	Yes
2019	[31]	U.S.A.	F	64	Local	Skin, limbs	*Alternaria alternata*	Culture	PZL	24	Yes	Heart	FR	10	None
2016	[32]	U.S.A.	F	65	Local	Skin, arm	*Biatriospora mackinnonii*	Culture + molecular biology (rDNA D1-D2)	ITZ	24	Yes	Kidney	FR	2	None
2011	[33]	U.S.A.	M	55	Disseminated	Brain	*Bipolaris spicifera*	Histopathology + direct microscopy	LAB + VZL	48	No	Heart	FR	1.5	Yes
2021	[34]	U.S.A.	M	69	Disseminated	Lung and brain	*Cladophialophora bantiana*	Culture	ISZ + LAB	-	No	Kidney	DFI	36	None
2017	[35]	U.S.A.	M	77	Local deep	Skin, leg, and feet	*Exophiala oligosperma*	Culture	ITZ	20	Yes	Kidney	FR	-	Yes
2019	[36]	U.S.A.	F	65	Local deep	Skin, foot	*Medicopsis romeroi*	Culture + molecular biology (ITS 1 + ITS 2)	PZL	12	Yes	Kidney	FR	70	Yes
2020	[37]	U.S.A.	M	64	Local	Skin, leg	*Nigrograna mackinnonii*	Culture + molecular biology (ITS2 + 28s-rRNA)	PZL	3	No	Kidney	FR	7	None
2012	[38]	U.S.A.	M	49	Local	Skin, leg	*Paraconiothyrium cyclothyrioides*	Culture + molecular biology	VZL, PZL	12	No	Kidney	FR	18	None
2014	[39]	U.S.A.	F	49	Disseminated	Lung	*Phaeoacremonium parasiticum*	Culture + direct microscopy	PZL	16	No	Kidney	FR	72	None
2022	[40]	U.S.A.	M	71	Local	Skin, hand	*Phialophora* spp.	Culture + molecular biology (ITS)	ITZ	-	No	Kidney	FR	16	None
2020	[41]	U.S.A.	M	40	Local	Skin, hand	*Rhytidhysteron rufulum*	Culture + molecular biology	VZL	12	Yes	Kidney	FR	-	None

PZL = posaconazole. VZL = voriconazole. ITZ = itraconazole. LAB = liposomal amphotericin B. ISZ = isavuconazole. ADF = anidulafungin. CSF = caspofungin. TBF = terbinafine. FTS = flucytosine. FR = full recovery. DOR = death for other reason. DFI = death for fungal infection. ITS = internal transcribed spacer.

**Table 5 jof-09-00283-t005:** Reported cases in South America.

Year	Ref	Country	Sex	Age	Dissemination	Localization	Species	Method of Identification	Therapy	Duration of Therapy in Weeks	Surgery	Transplant	Outcome	Time Tx-Diagnosis in Months	Susceptibility
2014	[42]	Brazil	M	68	Local	Skin, hand	*Alternaria infectoria, Colletotrichum gloeosporioides*	Culture	Surgery	-	Yes	Kidney	FR	35	Yes
2019	[43]	Colombia	M	66	Disseminated	Brain and lung	*Alternaria* spp.	Culture + direct microscopy	LAB + TBF	16	No	Kidney	DOR	12	None
2021	[44]	Brazil	F	46	Local	Skin, face	*Biatriospora mackinnonii*	Culture + molecular biology (ITS 4 + ITS 5)	ITZ	-	Yes	Kidney	FR	96	None
2019	[45]	Colombia	-	-	Disseminated	Brain	*Cladophialophora bantiana*	Culture+ direct microscopy	VZL	-	Yes	Kidney	FR	60	None
2016	[46]	Brazil	M	53	Local	Skin, leg	*Exophiala* spp.	Culture	ITZ	20	Yes	Kidney	FR	48	None
2016	[46]	Brazil	M	59	Local	Skin, leg	*Exophiala* spp.	Culture	ITZ	16	Yes	Kidney	FR	16	None
2016	[47]	Brazil	M	54	Local	Skin, leg	*Exophiala bergeri*	Culture + molecular biology	ITZ	-	No	Kidney	FR	24	None
2019	[48]	Brazil	M	45	Local	Skin, leg	*Exophiala xenobiotica*	Culture + molecular biology (ITS 1 + ITS 4)	ITZ	12	Yes	Kidney	FR	21	Yes
2016	[47]	Brazil	M	75	Local	Skin, limbs	*Exophiala xenobiotica*	Culture + molecular biology	ITZ	-	No	Kidney	Lost on follow-up	12	None
2016	[47]	Brazil	M	43	Local	Skin, hand	*Fonsecaea monophora*	Culture + molecular biology	Surgery	-	Yes	Kidney	FR	24	None
2016	[47]	Brazil	M	60	Local	Skin, arm	*Fonsecaea pedrosoi*	Culture + molecular biology	ITZ + surgery	-	Yes	Kidney	FR	36	None
2016	[47]	Brazil	M	57	Local	Skin, arm	*Fonsecaea spp.*	Culture + molecular biology	TBF	-	No	Kidney	FR	1	None
2017	[49]	Argentina	M	48	Local	Skin, hand	*Graphium basitruncatum*	Culture + molecular biology	VZL + surgery	-	Yes	Heart	FR	48	None
2015	[50]	French Antilles	M	71	Disseminated	Skin, lung, and brain	*Phaeoacremonium parasiticum and Paraconiothyrium cyclothyrioides*	Culture + molecular biology	VZL + LAB	-	Yes	Kidney	DOR	12	Yes
2014	[51]	French Antilles	F	59	Local	Skin, leg	*Pleurostoma ootheca*	Culture + direct microscopy	PZL	-	No	Kidney	FR	168	None
2016	[47]	Brazil	F	59	Local	Skin, foot	*Undefined*	Culture + molecular biology	ITZ + surgery	-	Yes	Kidney	FR	14	None

PZL = posaconazole. VZL = voriconazole. ITZ = itraconazole. LAB = liposomal amphotericin B. ISZ = isavuconazole. ADF = anidulafungin. CSF = caspofungin. TBF = terbinafine. FTS = flucytosine. FR = full recovery. DOR = death for other reason. DFI = death for fungal infection. ITS = internal transcribed spacer.

**Table 6 jof-09-00283-t006:** Reported cases in Asia.

Year	Ref	Country	Sex	Age	Dissemination	Localization	Species	Method of Identification	Therapy	Duration of Therapy in Weeks	Surgery	Transplant	Outcome	Time Tx-Diagnosis in Months	Susceptibility Test
2015	[52]	China	M	61	Local	Skin, leg	*Alternaria species*	Culture + direct microscopy	ITZ + VZL	7	No	Kidney	DOR	12	None
2020	[53]	Japan	F	40	Local	Skin, leg	*Exophiala jeanselmei*	Culture + direct microscopy	ITZ	92	Yes	Kidney	FR	72	None
2017	[54]	India	-	16	Local	Skin, leg	*Exophiala jeanselmei*	Culture	ITZ	12	Yes	Kidney	FR	11	None
2016	[55]	India	F	48	Local	Skin, leg	*Exophiala jeanselmei*	Culture + direct microscopy	LAB + ITZ	6	Yes	Kidney	FR	6	None
2012	[56]	China	M	66	Local	Skin, arm	*Exophiala jeanselmei*	Molecular biology (ITS sequence)	Surgery + local injection and undefined systemic antifungal treatment	-	Yes	Kidney	FR	67	Yes
2020	[57]	China	F	47	Local	Skin, hand	*Hongkongmyces snookiorum*	Culture + molecular biology (ITS 1 + ITS 2)	VZL	8	Yes	Kidney	FR	18	Yes
2017	[54]	India		37	Local	Skin, foot	*Neoscytali* *dium species*	Culture	ITZ	24	Yes	Kidney	FR	1	None
2015	[58]	Japan	F	61	Local	Skin, arm	*Phaeoacremonium* spp.	Culture + direct microscopy	LAB	2	No	Kidney	DOR	156	None
2021	[59]	China	M	65	Disseminated	Liver	*Pleurostoma hongkongense sp. nov.*	Culture + molecular biology (ITS + 28s nr-DNA + 18s nr-DNA)	ADF, LAB, VZL	14	Yes	Liver	FR	16	Yes
2019	[60]	Thailand	M	57	Disseminated	Liver	*Pleurostomophora richardsiae*	Culture + molecular biology (ITS 1 + ITS 2)	LAB	4	Yes	Liver	FR	1	Yes
2019	[61]	Singapore	M	-	Local	Skin, leg	*Pleurostomophora richardsiae*	Culture + direct microscopy	ITZ	40	Yes	Kidney	FR	276	None
2017	[62]	India	F	43	Local	Skin, leg	*Pyrenochaeta romeroi*	Culture + direct microscopy	ITZ + TBF, VZL	8	Yes	Kidney	FR	6	None
2021	[63]	Kuwait	F	50	Disseminated	Liver, brain, and lung	*Rhinocladiella mackenziei*	Culture + molecular biology (ITS)	LAB + VZL	6	No	Kidney	DFI	3	None
2013	[64]	Thailand		69	Disseminated	Brain	*Scedosporium apiospermum and Phaeoacremonium parasiticum*	Culture + direct microscopy	VZL	24	No	Kidney	PR	168	None
2017	[54]	India		21	Local	Skin, foot	*Undefined*	Culture	ITZ	72	Yes	Kidney	FR	6	None
2017	[54]	India		22	Local	Skin, hand	*Undefined*	Culture	ITZ	68	Yes	Kidney	FR	6	None
2017	[54]	India	F	49	Local	Skin, arm	*Undefined*	Culture	ITZ	48	Yes	Kidney	DOR	12	None
2017	[54]	India		43	Local	Skin, foot	*Undefined*	Culture	Surgery	-	Yes	Kidney	FR	6	None
2017	[54]	India		23	Local	Skin, foot	*Undefined*	Culture	ITZ	12	Yes	Kidney	FR	5	None
2022	[65]	India	F	36	Disseminated	Skin and bone	*Undefined*	Direct microscopy	Undefined	-	Yes	Kidney	FR	72	None
2021	[66]	India	M	50	Local	Skin, foot, and leg	*Undefined*	Culture + direct microscopy	ITZ	-	Yes	Kidney	PR	24	None
2021	[66]	India	M	55	Local	Skin, hand, and foot	*Undefined*	Culture + direct microscopy	VZL + CSF	-	Yes	Kidney	DOR	3	None
2021	[66]	India	M	35	Disseminated	Skin, foot, and peri-renal abscess	*Undefined*	Culture + direct microscopy	LAB	-	Yes	Kidney	FR	24	None
2021	[66]	India	M	52	Local deep	Facial skin and bone	*Undefined*	Culture + direct microscopy	LAB	-	Yes	Kidney	FR	17	None
2021	[66]	India	M	52	Local	Skin, foot	*Undefined*	Culture + direct microscopy	LAB	-	Yes	Kidney	FR	1	None
2016	[67]	India	M	37	Local deep	Salivary gland	*Undefined dematiaceous fungi*	Direct microscopy	VZL	-	Yes	Kidney	FR	39	None
2016	[67]	India	M	37	Local deep	Salivary gland	*Undefined dematiaceous fungi*	Direct microscopy	VZL	-	Yes	Kidney	FR	39	None

PZL = posaconazole. VZL = voriconazole. ITZ = itraconazole. LAB = liposomal amphotericin B. ISZ = isavuconazole. ADF = anidulafungin. CSF = caspofungin. TBF = terbinafine. FTS = flucytosine. FR = full recovery. DOR = death for other reason. DFI = death for fungal infection. ITS = internal transcribed spacer.

**Table 7 jof-09-00283-t007:** Reported cases in Australia.

Year	Ref	Country	Sex	Age	Dissemination	Localization	Species	Method of Identification	Therapy	Duration of Therapy in Weeks	Surgery	Transplant	Outcome	Time Tx-Diagnosis in Months	Susceptibility Test
2015	[68]	Australia	F	49	Local	Skin, leg	*Microsphaeropsis arundinis*	Culture + molecular biology	VZL	24	No	Kidney	FR	84	Yes
2015	[68]	Australia	F	70	Local	Skin, leg, and arm	*Microsphaeropsis arundinis*	Culture + molecular biology	LAB + PZL	48	Yes	Kidney	FR	4	Yes
2015	[68]	Australia	M	55	Local	Skin, arm	*Microsphaeropsis arundinis*	Culture + molecular biology	ITZ	44	Yes	Kidney	FR	30	Yes
2016	[69]	Australia	F	67	Disseminated	Heart and brain	*Verruconis gallopava*	Molecular biology	VZL + ADF	-	No	Kidney	DFI	18	Yes

PZL = posaconazole. VZL = voriconazole. ITZ = itraconazole. LAB = liposomal amphotericin B. ISZ = isavuconazole. ADF = anidulafungin. CSF = caspofungin. TBF = terbinafine. FTS = flucytosine. FR = full recovery. DOR = death for other reason. DFI = death for fungal infection. ITS = internal transcribed spacer.

## 4. Discussion

Phaeohyphomycosis (from the Greek word “phaeo” meaning “dark”) is caused by pigmented dematiaceous fungi and can cause a wide spectrum of clinical diseases, ranging from superficial to disseminated infections [6]. Immunocompromised hosts, including SOT recipients, are the most affected category of patients [35]. It is difficult to accurately determine the incidence of phaeohyphomycosis in the SOT population because of the rarity of the disease, and previous studies are limited to case reports and retrospective studies [3]. We report three cases of phaeohyphomycosis local infections in SOT recipients caused by *Alternaria alternata, Alternaria infectoria*, and *Curvularia hawaiiensis*. In our cases, the length of therapy and choice of intervention (surgery, antifungals, or both) for each clinical entity were based primarily on the clinical presentation, the underlying condition of the host, and the initial response. In the two heart recipients (case 1 and 2), because of a more intense immunosuppression and recent treatment of acute rejection, surgical excision with systemic antifungal therapy was chosen. The kidney transplant recipient (case 3) requiring a less intense immunosuppressive regimen, surgical excision, without antifungal treatment, and follow-up was considered to be sufficient. All three patients survived, with full recovery.

Though phaeohyphomycosis is a rare fungal infection, trends toward an increasing incidence have been noted. In our review, most of the cases (75%) were published in the last seven years, confirming the increasing trend observed in recent years and already described by Schieffelin et al. and by McCarthy [3,70]. This is likely a consequence of medical advancements allowing for increased transplantation rates, resulting in increasing numbers of immunosuppressed patients who are at risk for such opportunistic infections. Interestingly, we additionally found substantial differences in characteristics of affected patients and causative organisms between cases reported in the EU and the rest of the world. While kidney transplant recipients were most frequently reported to be diagnosed with phaeohyphomycosis in every country, surprisingly, lung transplant recipients were only reported in the EU. This is inconsistent with what was reported by the U.S.-based (Transplant-Associated Infection Surveillance Network) TRANSNET cohort and Schieffelin at al. cohort, where 53% and 11% of the SOT recipients with phaeohyphomycosis were lung recipients, respectively [3,70]. This finding can be partially explained by the different time periods of the cases collection. We reviewed cases published from 2011 to 2022, Schieffelin at al. reported cases from 1988 to 2009, while in the TRANSNET cohort, cases were reported from 2001 to 2006. Regarding the time of onset of the infection, we found a median time from transplant to infection of 18 months; a similar time range was observed in the 2 cohorts previously mentioned. In our review, most of the infections described were localized to the skin and soft tissue. In fact, only 22% of them were disseminated to other organs. This finding is discordant with what is described in the TRANSNET cohort, where 63.3% of the infections were disseminated [3]. This can partly explain the higher mortality observed in the TRANSNET cohort since disseminated infections are more severe than local ones.

Our literature review revealed about 40 different species of fungi causing phaeohyphomycosis; Alternaria and Exophiala resulted to be the most reported genera. Previous studies have shown a similar distribution of genera, with Alternaria and Exophiala being the most common genera involved; however, fewer species were identified [3,70]. The increasing number of pathogens could be attributable to a wider use of molecular biology, which provides a more accurate species identification including unusual pathogens. In addition to species identification, antifungal susceptibility testing should be performed with a microscopic examination and culture. Despite the fact that the guidelines of the European Society of Clinical Microbiology and Infectious Diseases (ESCMID) guidelines [6] as well as the American Society of Transplantation Infectious Diseases Community of Practice (AST-IDCOP) [71] suggest obtaining a susceptibility test, in our review, we found that it was carried out only in about a third of cases (28% of cases in the EU vs. 26% in non-EU). Antifungal susceptibility testing could be useful to guide antifungal therapy; however, a correlation between in vitro MICs and patient outcomes is still debated due to the scarcity of robust clinical data. This is especially the case for rare fungal pathogens [72]. Therefore, the majority of therapies reported were empirical.

There are no standardized therapies for phaeohyphomycosis. ESCMID guidelines [6] suggest voriconazole, posaconazole, and itraconazole as empirical therapy. These suggestions are reflected in the European cases of our review, in which voriconazole was widely used, followed by itraconazole and posaconazole. In the TRANSNET cohort, the most used antifungal agent was voriconazole (44%), while in Schieffelin et al.’s review, itraconazole was the most used agent for skin lesions, but a combination of voriconazole and amphotericin B was preferred for disseminated disease. Surgical resection was performed in 67% of the European patients, with 17% of them not receiving an antifungal therapy. Similarly, in non-EU countries, surgery was performed in 69% of cases, but only 5% did not receive antifungals. In Schieffelin et al.’s cohort, most of the patients received surgery (81%), not followed by antifungal therapy in 14% of the cases. Phaeohyphomycosis mortality is related to the dissemination of the disease. The mortality rate was 5% overall compared to 47% mortality in the TRANSNET cohort, probably due to the greater proportion of disseminated disease.

To the best of our knowledge, this is the largest review of the literature on phaeohyphomycosis in SOT recipients. These infections have not been studied in clinical trials and the available therapeutic data are primarily based on sporadic case reports. This review additionally highlights that a standardized approach to phaeohyphomycosis is difficult to determine due to the wide spectrum of dematiaceous fungi involved, different characteristics of the hosts, and the variety of clinical presentations. An early diagnosis and therapy are critical in preventing the dissemination of disease; therefore, in SOT recipients with atypical cutaneous lesions, a skin biopsy should always be performed. The diagnosis relies on a high index of clinical suspicion paired with an accurate mycological investigation. In addition to a histologic examination, culture and molecular biology identification are essential to establish an etiologic diagnosis in rare fungal diseases. As cases increase, further studies are becoming necessary to determine the optimal management strategy in this vulnerable immunosuppressed population.

## Figures and Tables

**Figure 1 jof-09-00283-f001:**
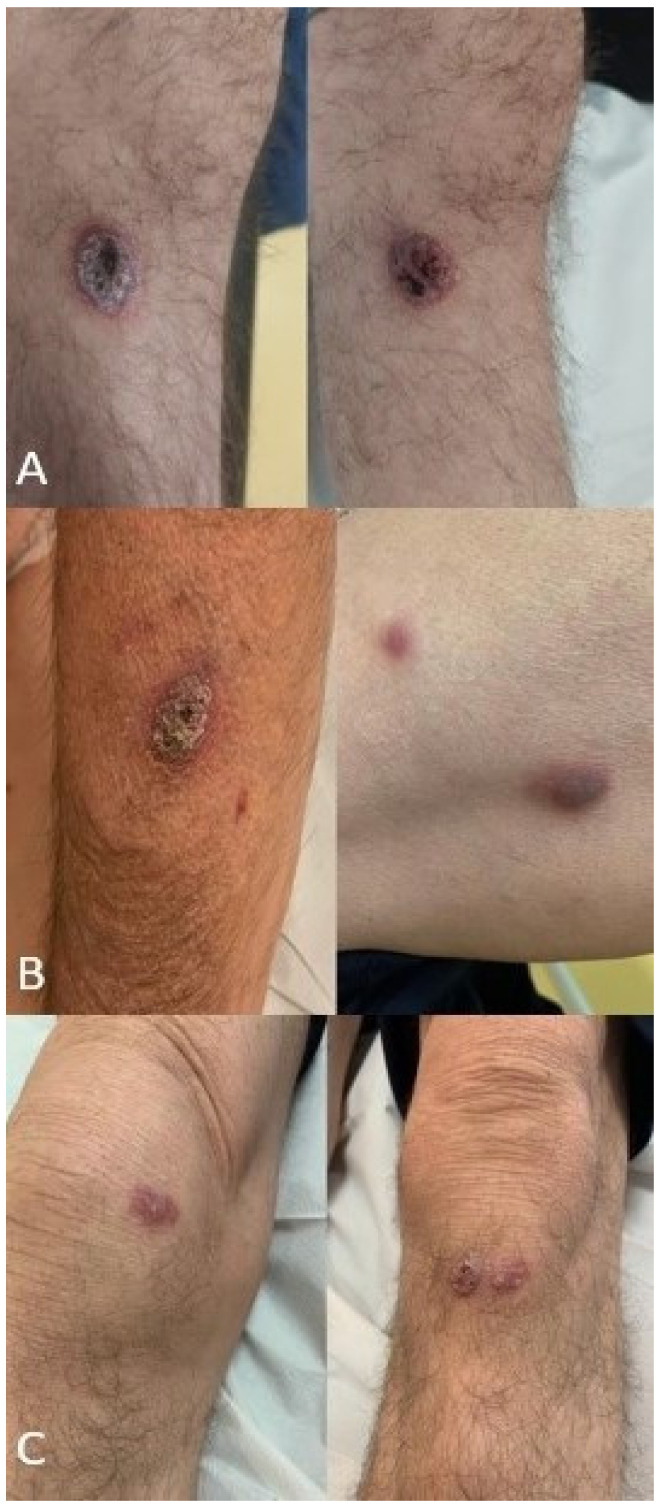
Clinical presentation of cutaneous phaeohyphomycosis. (**A**) *Curvularia hawaiiensis*; (**B**). *Alternaria alternata*; (**C**). *Alternaria infectoria*.

**Figure 2 jof-09-00283-f002:**
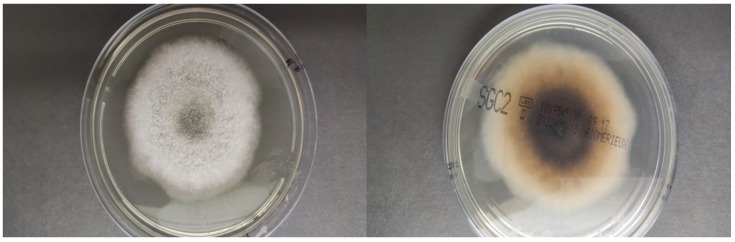
*Alternaria alternata* culture, the two sides of the plate.

**Figure 3 jof-09-00283-f003:**
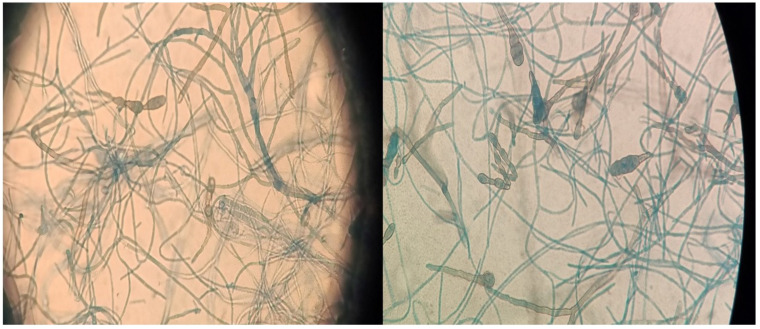
*Alternaria alternata* microscopy.

**Table 1 jof-09-00283-t001:** Susceptibility test of the isolates.

Antifungal Agent	*Curvularia hawaiiensis* (mg/L)	*Alternaria alternata* (mg/L)
Amphotericin B	4	1
Anidulafungin	0.016	0.03
Micafungin	0.016	0.03
Caspofungin	0.016	0.125
Isavuconazole	0.25	1
Posaconazole	0.016	0.06
Voriconazole	0.003	1
Itraconazole	0.003	0.25
Fluconazole	4	16

MIC: minimum inhibitory concentration; mg/L: milligrams/liter.

**Table 2 jof-09-00283-t002:** Summarized results of the literature review.

	EU (*n* = 36)	Non-EU (*n* = 58)	Total (*n* = 94)
Age at disease presentation	58.50 [49.50–65.75]	54.00 [45.25–64.75]	56.0 [47.25–65.00]
Transplant to diagnosis time	18.00 [8.75–48.00]	16.50 [6.00–45.75]	18.00 [7.00–48.00]
Transplant Organ			
Kidney	16 (44.4%)	53 (91.4%)	69 (73.4%)
Lung	13 (36.1%)	0 (0.0%)	13 (13.8%)
Heart	4 (11.1%)	3 (5.2%)	7 (7.4%)
Liver	3 (8.3%)	2 (3.4%)	5 (5.5%)
Sex, M	28 (77.7%)	31 (53.4%)	59 (62.8%)
Dissemination			
Local	26 (72.2%)	40 (70.0%)	66 (70.2%)
Local deep	3 (8.3%)	4 (6.90%)	7 (7.4%)
Disseminated	7 (19.4%)	14 (24.1%)	21 (22.3%)
Genus			
* Alternaria*	16 (44.4%)	4 (6.9%)	20 (21.2%)
* Exophiala*	3 (8.3%)	10 (17.2%)	13 (13.8%)
* Cladiophialophora*	3 (8.3%)	2 (3.4%)	5 (5.3%)
* Medicopsis*	3 (8.3%)	1 (1.7%)	4 (4.3%)
Other	11 (30.6%)	31 (53.5%) *	42 (44.7%) *
Not identified	0 (0.0%)	13 (22.4%)	13 (13.8%)
Species identification			
Molecular biology	28 (77.8%)	26 (44.8%)	54 (57.4%)
Microscopy or histology	6 (16.7%)	19 (32.8%)	25 (26.6%)
Not done/reported	4 (11.1%)	13 (22.4%)	17 (18.1%)
Susceptibility testing done	10 (27.8%)	15 (25.9%)	25 (26.6%)
Treatment			
Antifungal alone	12 (33.3%)	18 (31.0%)	30 (31.9%)
Surgery alone	6 (16.7%)	3 (5.2%)	9 (9.6%)
Surgery + antifungal	18 (50%)	37 (63.8%)	55 (58.5%)
Clinical outcome			
Recovery	29 (80.6%)	48 (82.8%)	77 (81.9%)

* In three cases, two microorganisms were identified.

## Data Availability

Data supporting the case report are available upon request to the corresponding author.

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
