# Peer review of "Phaeohyphomycosis in Solid Organ Transplant Recipients: A Case Series and Narrative Review of the Literature"

_jof, 2023, doi:10.3390/jof9030283_

Round 1

Reviewer 1 Report

This manuscript presents three case studies due to once rare fungi followed by a literature review. In total it is a nice composite study of the current status of phaeohyphomyocosis in SOT recipients.

A few comments

On line 79, Figure 1 B is referred to - I think this should be Figure 1 A

In figures 2 and 3, please state what the difference between the two plates. Two different isolates/ isolates from different lesions?

In line 250 it is stated that there is an inconsistency in the TRANSNET report and the author's findings. Did the authors check the sources in the TRANSNET report - is it based on unpublished cases?

Finally it would be interesting to note if the rise in incidence of phaeohyphomycosis in SOT patients is consistent in rise of these infections in the immunocompromised population or if they are 'specific' for SOT.

Reviewer 2 Report

In this case report and narrative review, the authors describe 3 solid organ transplant recipients who had infections with phaehyphomycosis. They reviewed the literature and report on 94 other cases in transplant recipients.

Overall, the authors provide a detailed and extensive review of this emerging group of fungal infections and their case report provides a comprehensive overview of it. There are certain points I wish to raise below that will help improve the educative value of this study.

Major Comments

1. In patient #1, with the findings of ptosis and the pulmonary nodules, was there concern for Horner’s Syndrome? Was the ptosis and pulmonary nodules thought to be related to the underlying fungal infection? It would be helpful if the authors clarify this point, particularly as the pulmonary lesions were unchanged 2 months after therapy.

2. The study authors report susceptibility testing of the fungal isolates in their study. It would be of educative value if they mentioned which guidelines were used in interpretation – was it the 2014 ESCMID and ECMM joint clinical guidelines (https://www.clinicalmicrobiologyandinfection.com/article/S1198-743X(14)60230-5/fulltext#seccestitle50)? Additionally, for the other cases reported to have susceptibility testing performed, did they make use of the same susceptibility guidelines and breakpoints or were there other testing methods utilized?

3. In Tables 4-7, the study authors note outcomes such as “Full Recovery” and “Recovery”, as well as certain patients with outcomes documented as “PR”. Does PR refer to “Partial recovery”? What were the criteria to denote patient as “fully recovered” versus “recovered” versus “partially recovered”? It would be helpful to include those points in the Materials and Methods section.

4. The authors note that in their review, some of their findings were discordant with the TRANSNET study and with the Schieffelin et al. study – primarily the distribution of lung transplant recipients with phaehyphomycosis and the percentage of patients with disseminated infections. Could the authors comment on why these particular differences were seen? One particular finding could be related to the changing fungal epidemiology over the past few years, but it would be of interest to the readers if this was explored further.

Minor Comments

1. Page 3 of 26, lines 125-127 – Patient #2 had subtherapeutic itraconazole levels and was asked to change the way he took his other medications. Was there repeat Itraconazole levels checked later in the course of therapy for him?

2. Table 7, pages 20 of 26 – there appear to be minor errors in editing for entries related to reference (63) and the first patient reported in (64), especially in relation to type of organ transplanted and outcomes. Please rectify those.

3. The Discussion raises some extremely useful talking points, but the volume of information contained in a single paragraph (especially lines 225-285) makes it difficult to read. It would be helpful, if the study authors consider dividing this into smaller paragraphs.

4. The page numbers of the section beginning from the Discussion appears to be incorrect. Please rectify this.
